# Direct 3-D Sparse Imaging Using Non-Uniform Samples Without Data Interpolation

**Dou Sun [1], Bo Pang [1], Shiqi Xing [1,\*], Yongzhen Li [1] and Xuesong Wang [2]**

[1] State Key Laboratory of Complex Electromagnetic Environment Effects on Electronics and Information System, National University of Defense Technology, Changsha 410073, China; sundou14@nudt.edu.cn (D.S.); pangbo84826@126.com (B.P.); e0061@sina.com (Y.L.)

[2] College of Electronic Science, National University of Defense Technology, Changsha 410073, China; wxs1019@vip.sina.com

\* Correspondence: xingshiqi_paper@163.com; Tel.: +86-186-7072-9869

**Abstract:** As an emerging technique, sparse imaging from three-dimensional (3-D) and non-uniform samples provides an attractive approach to obtain high resolution 3-D images along with great convenience in data acquisition, especially in the case of targets consisting of strong isolated scatterers. Although data interpolation in k-space and fast Fourier transform have been employed in the existing 3-D sparse imaging methods to reduce the computational complexity, the data-gridding errors induced by local interpolation may usually result in poor imaging performance. In this paper, we directly regard the imaging problem as a joint sparse reconstruction problem from non-uniform data without interpolation in 3-D space. Combining dictionary reduction and Gauss iterative method with the optimized signal processing scheme, a sparse imaging algorithm is proposed to address the difficulty of large-scale computation involved in direct 3-D sparse reconstruction. Benefited from the optimized signal processing scheme and the avoidance of data interpolation, the direct 3-D sparse imaging (DTDSI) method proposed in this paper is of low computation scale and high imaging performance. Experiments of electromagnetic simulation data demonstrate the DTDSI method outperforms baseline methods in terms of resolving ability, lower side-lobes and higher accuracy.

**Keywords:** three-dimensional (3-D); sparse imaging; non-uniform; data interpolation

## 1. Introduction

In the development of synthetic aperture radar (SAR) technology, extensive efforts have been made to acquire high-resolution three-dimensional (3-D) imaging results [1,2]. Generally, uniform and dense samples in 3-D k-space can be processed by tomographic SAR (Tomo-SAR) [3,4] or holographic SAR (Holo-SAR) [5,6] to produce high-resolution 3-D imaging results. Generating high-resolution 3-D images using uniform and dense samples requires that the data be collected over a densely sampled set of points in both azimuth and elevation angle [7], as shown in the cyan dotted lines representing a scanning trajectory of Tomo-SAR along the elevation and the azimuth in Figure 1. Collecting data from many closely spaced linear flight passes need the targets be observed by the SAR system several times, thus making data collection time-consuming and high-cost [7,8]. In addition, acquiring uniform and dense samples is impractical in several applications, such as military surveillance. Consequently, there is motivation to consider the imaging using non-uniform samples, which are more fundamental and critical than uniform and dense samples in practical applications.

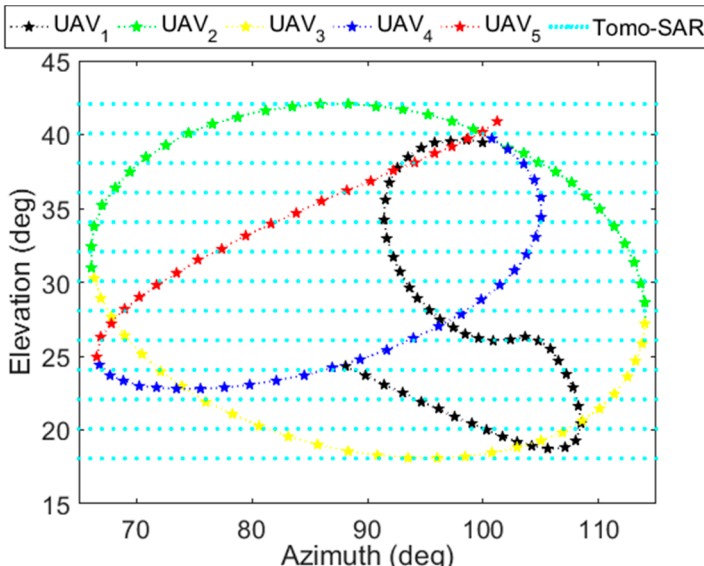

**Figure 1.** The scanning trajectories of Tomo-SAR (tomographic SAR) and five UAVs (unmanned aerial vehicles) with SAR systems in cooperative flights.

As a novel technology, imaging from 3-D and non-uniform samples brings great convenience in data acquisition to obtain high resolution 3-D imagery [9–11], especially in the case of targets consisting of strong isolated scatterers. A feasible way to collect 3-D and non-uniform samples is through cooperative flights of multiple small unmanned aerial vehicles (UAVs) with SAR systems [12–14]. The curve trajectory in [7] is used as the trajectory for UAVs. As shown in Figure 1, the green, red, blue, yellow and black curves represent the scanning trajectory of five UAVs along non-linear flight paths, respectively. It is obvious that the scanning trajectories of five UAVs are much sparser than the scanning trajectory of Tomo-SAR that collects dense samples. Due to the sparse data collection and good maneuverability of UAV platforms, collecting non-uniform samples saves cost greatly and is feasible for several practical applications, i.e. military surveillance. However, since the azimuth and the elevation are coupling and a complex baseline is formed in non-uniform samples, traditional 3-D imaging methods [5,15,16], which are generally used to solve the imaging problem for uniform and dense samples, can hardly be suitable for 3-D SAR using non-uniform samples. The imaging for non-uniform samples, what we pay attention on in this paper, is very different from the imaging for uniform and dense samples.

In the 3-D imaging literature, there are few published works focusing on non-uniform samples, and the involved difficulties rely on the fact that linear filtering is not applicable for imaging since the k-space samples of 3-D SAR using non-uniform samples is sparse and non-uniform. Specifically, due to the high coupling between the elevation and the azimuth resulting from the non-uniform k-space samples, it is impossible to estimate the height independently after the two-dimensional imaging [17]. Although full 3-D imaging is an advisable way, using Fourier processing methods, such as 3-D non-uniform fast Fourier transform (3-D NUFFT), generates poor imaging results with high side-lobes due to the sparsity of the k-space samples [17,18]. Sparse reconstruction is a method of model matching, in which regularization enforcing sparsity to obtain sparse results [19,20]. Combining the sparse reconstruction with the sparsity of the data to carry out 3-D sparse imaging for non-uniform samples is expected to reduce high side-lobes and obtain high-resolution imaging results. However, the large-scale computation involved in 3-D sparse reconstruction makes it a challenging and difficult task [21]. To reduce the computational complexity, Austin proposed a sparse imaging method based on data interpolation in k-space and fast Fourier transform (FFT) operation [7]. After interpolation, the updated dictionary matrix can be replaced by FFT operation for fast calculation in the procedure of sparse reconstruction due to the uniformity of the interpolated data. Although this method solves

the difficulty of large-scale computation and makes sparse imaging for non-uniform samples feasible, the data interpolation by local information involves the high potential for data-gridding errors and in turn affect the accuracy of the imaging results.

In order to exploit sparse imaging and avoid the adverse effects caused by local interpolation, a direct 3-D sparse imaging (DTDSI) method is proposed in this paper. Firstly, the imaging problem is directly regarded as a joint sparse reconstruction problem from non-uniform data without data interpolation in 3-D space. To address the difficulty of large-scale computation involved in direct 3-D sparse reconstruction, we then reduce the dimension of the dictionary matrix via candidate scattering centers selection. Finally, combining the Gauss iterative method and the optimized signal processing scheme, an algorithm is proposed to solve the updated sparse imaging model. To evaluate the performance of the DTDSI method, we compare it with 3-D NUFFT and Austin's method via experiments of electromagnetic simulation data. The experiments are conducted in MATLAB R2016b, and tested on a computer equipped with an Intel Core I5-6500 CPU and 12 GB RAM.

## 2. Methods and Data

### 2.1. Signal Model

We assume that the radar is far enough from the scene so that the plane wave model can be utilized. A radar, located at azimuth $\phi$ and elevation $\theta$ with respect to the scene center, transmits a wideband signal, the received signal can be expressed as:

$$r(t;\phi,\theta) = \left[ \iint g(x = \frac{ct}{2}, y, z; \phi, \theta) dy dz \right] * s(t) \tag{1}$$

where $t$ is the time, $c$ is the speed of light, $s(t)$ is the known wideband signal with center frequency $f_c$ and bandwidth $BW$, $*$ represents convolution. The reflectivity function of the scene is given by $g(x, y, z; \phi, \theta)$, where $x, y, z$ represents the position of the target in the scene.

Equation (1) can be understood as the Fourier transform of the scene reflectivity function projected onto the *x–dimension*. According to the projection slice theorem, the reflectivity function $g(x, y, z; \phi, \theta)$ satisfies the following formula:

$$G(k_x, k_y, k_z) = \int g(x, y, z; \phi, \theta) \cdot e^{-j(k_x x + k_y y + k_z z)} dx dy dz \tag{2}$$

where $G(k_x, k_y, k_z)$ is the k-space measurements obtained from the received signal $r(t; \phi, \theta)$ and the wideband signal $s(t)$.

The support of each k-space measurement is a line segment in k-space samples $(k_x, k_y, k_z)$ with extent $4\pi BW/c$ rad/m centered at $4\pi f_c/c$ rad/m, and oriented at observation angle $(\theta, \phi)$ determined by the location of the radar. A set of k-space samples indexed on $(j, q)$ is given by:

$$\begin{aligned}
k_x^{j,q} &= \frac{4\pi f_j}{c} \cos \theta_q \cos \phi_q \\
k_y^{j,q} &= \frac{4\pi f_j}{c} \cos \theta_q \sin \phi_q \\
k_z^{j,q} &= \frac{4\pi f_j}{c} \sin \theta_q
\end{aligned} \tag{3}$$

where the frequency $f$ is sampled as $f \to f_j$ and the observation angle is sampled as $(\theta, \phi) \to (\theta_q, \phi_q)$.

The task of this paper is to estimate the reflectivity function $g(x, y, z; \phi, \theta)$ from the known k-space measurements $(k_x, k_y, k_z)$ according to their relationship shown in Equation (2). For the 3-D SAR using non-uniform samples, the scanning trajectory of radar (see Figure 1) is a set of random curves, which leads to the formation of a complex baseline. Consequently, the k-space samples $G(k_x, k_y, k_z)$, which are determined by the observation angle $(\theta, \phi)$, namely, the scanning trajectory, are non-uniform and sparse, as shown in Figure 2. In order to obtain 3-D high-resolution imaging results for the

non-uniform and sparse k-space samples, we combine the sparse reconstruction with the sparsity of the data and propose the DTDSI method.

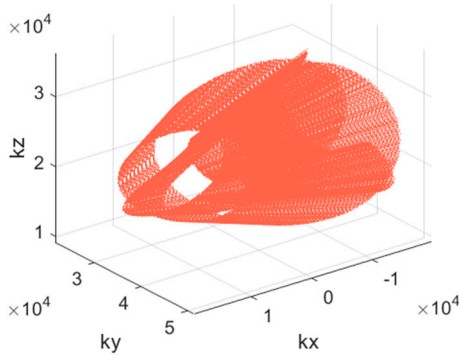

**Figure 2.** The k-space samples of five UAVs with SAR system in cooperative flight.

### 2.2. Direct 3-D Sparse Imaging Modeling

Generally, the underlying scene of SAR image is sparse, which consists of only a small number of dominate strong scatterers, especially for man-made objects [15]. Considering the sparsity of the scene, modelling the imaging problem as a sparse reconstruction problem is expected to improve the quality of imaging results.

In image reconstruction space, a set of $N$ locations are defined as candidate scattering centers:

$$C = \{(x_n, y_n, z_n)\}_{n=1}^N \tag{4}$$

Usually, these locations are selected from a uniform rectangular grid. Based on these locations and the k-space samples, the $M \times N$ − dimensional dictionary matrix is given by:

$$A = \left[ e^{-j(k_{x,m}x_n + k_{y,m}y_n + k_{z,m}z_n)} \right]_{m,n} \tag{5}$$

where $m = j * q$ indicates the index of the $M$ measurements in k-space and $n$ indicates the index of the $N$ locations in $C$.

In order to avoid errors caused by data interpolation, we abandon data interpolation and directly utilize the dictionary matrix in Equation (5) to build the direct 3-D sparse imaging model.

Combining Equation (5), we rewrite Equation (2) into associated matrix form and obtain:

$$b = A\beta \tag{6}$$

where vector $\beta$ is the 3-D image we want to reconstruct and vector $b$ is $M$ − dimensional k-space measurements.

3-D sparse imaging is to solve the following sparse reconstruction problem:

$$\widetilde{\beta} = \underset{\beta}{\mathrm{argmin}} \|\beta\|_0 \text{ subject to } b = A\beta \tag{7}$$

where $\|\cdot\|_0$ represents $l_0 − norm$, $\|\beta\|_0$ denotes the number of nonzero elements in $\beta$ and $\widetilde{\beta}$ represents the reconstructed imaging results.

In fact, solving the direct 3-D sparse imaging model shown in Equation (7) faces the problem of large computation and storage pressure. Assume that the size of the scene we want to reconstruct is 10 m × 10 m × 10 m and the resolution of each dimension is 0.05 m, thus the size of the 3-D image space is $200 \times 200 \times 200$ with $N$ equals to $8 \times 10^6$. Furthermore, we assume that the number of samples in frequency and in observation angle are 200 and 600, respectively, thus $M = 1.2 \times 10^5$. Because the storage of the $M \times N$ − dimensional dictionary matrix $A$ needs 14,305 GB space, it is impossible

to store $A$. In addition, the calculation of $A\beta$, which is inevitable in sparse reconstruction algorithm, needs $M \times N = 9.6 \times 10^{11}$ multiplications per iteration. Faced with such large computation and storage pressure, it is unrealistic to directly apply the existing sparse reconstruction algorithms to solve Equation (7). To address the difficulty of large-scale computation involved in direct 3-D sparse reconstruction, a proper sparse imaging algorithm is needed.

*2.3. Proposed Sparse Imaging Algorithm*

Since the data interpolation is avoided in direct 3-D sparse imaging modeling, the authenticity of data is guaranteed. However, direct 3-D sparse reconstruction inevitably faces the difficulty of large-scale computation. To address this problem and get high-quality imaging results, an algorithm based on the dictionary reduction and the optimized signal processing scheme is proposed in this section.

In order to achieve the detection and identification of the target, we need to get the imaging result of the target by observing the scene. In the observation scene, only the target area is what we are interested in. And usually, the imaging results at most areas other than the target area are very small or even zero, especially in the case of target consisted of strong isolated scatterers. Therefore, once the imaging result at the target area is obtained, the imaging is complete. Based on this, the dictionary reduction by candidate scattering centers selection is carried out and the dimension of the sparse imaging model is reduced simultaneously.

To determine the location of the target area, we use other imaging methods, such as 3D-NUFFT, to get the initial imaging result $\widehat{\beta}$. In the initial imaging result $\widehat{\beta}$, the position with the amplitude greater than the threshold $\varepsilon$ belongs to the target area $C'$:

$$C' = \left\{ (x_n, y_n, z_n) \middle| \widehat{\beta}(x_n, y_n, z_n) > \varepsilon \right\}_{n=1}^N = \{(x_n', y_n', z_n')\}_{n=1}^{N1} \tag{8}$$

where $x_n', y_n', z_n'$ represent $N1$ candidate scattering centers in the target area. Corresponding to the region $C'$, $\beta'$ represents the imaging result at the target area and is what we need to reconstruct now. The relationship between $\beta'$ and $\beta$ satisfies:

$$\beta = \begin{cases} \beta' & (x, y, z) \in C' \\ 0 & others \end{cases} \tag{9}$$

After the selection of candidate scattering centers, the dictionary matrix is updated to $A'$:

$$A' = \left[ e^{-j(k_{x,m}x_n' + k_{y,m}y_n' + k_{z,m}z_n')} \right]_{m,n} \tag{10}$$

As the number of the selected candidate scattering centers is $N1$, the dimension of $A'$ has been reduced to $M \times N1$. Furthermore, the sparse imaging model is reduced to:

$$\widetilde{\beta'} = \underset{\beta'}{\mathrm{argmin}} \|\beta'\|_0 \text{ subject to } b = A'\beta' \tag{11}$$

Compared with Equation (7), the dimension of the updated sparse imaging model in Equation (11) is much too small. Benefitting from dictionary reduction, the computation and storage pressure are greatly reduced. After dictionary reduction, the sparse imaging problem becomes how solve Equation (11). The mathematical model in Equation (11) is not a convex optimization problem, but a NP-hard problem. Since $\ell_p - norm$ [22,23], which possesses the great capability of producing a sparser and more accurate solution, is very similar to $\ell_0 - norm$, we relax Equation (11) using $\ell_p - norm$ into the following optimization problem:

$$\widetilde{\beta'} = \underset{\beta'}{\mathrm{argmin}} \left( \|(b - A'\beta')\|_2^2 + \lambda \|\beta'\|_p^p \right) \tag{12}$$

where $\|\beta'\|_p^p = \sum_i \left( |\beta'(i)|^p \right)$, $\beta'(i)$ represents the $i-th$ element of $\beta'$, $p(0 < p \leq 1)$ represents the shrinkage parameter, $\lambda$ is the regularization parameter to control sparsity.

According to Equation (12), we define the cost function $J$:

$$J = \|(b - A'\beta')\|_2^2 + \lambda\|\beta'\|_p^p \tag{13}$$

When the cost function $J$ takes the minimum value, the result of sparse reconstruction can be obtained. Inspired by Cetin's processing idea [24], the partial derivative $J$ sub $\beta'$ is calculated to get the minimum value of $J$:

$$\nabla_{\beta'}J = (2A'^H A' + \lambda pD(\beta'))\beta - 2A'^H b \tag{14}$$

where $D(\beta') = diag\left( |\beta'(i)|^{p-2} \right)$ is a diagonal matrix of order *N1*.

Finding the minimum value of $J$ becomes the solving of $\nabla_{\beta'}J = 0$. When $\nabla_{\beta'}J = 0$, we will get the sparse reconstruction result, that is, the imaging result. Considering that $2A'^H A' + \lambda pD(\beta')$ can be approximately regarded as a Hessian matrix, the following approximate Gauss iteration method is used to solve $\nabla_{\beta'}J = 0$:

$$\widetilde{\beta}'_{k+1} = \widetilde{\beta}'_k - \Delta_{k+1}(2A'^H A' + \lambda pD(\widetilde{\beta}'_k))^{-1}\nabla_{\widetilde{\beta}'_k}J \tag{15}$$

where $\Delta_{k+1}$ is the step in each iteration, $\nabla_{\widetilde{\beta}'_k}J$ is the value of $\nabla_{\beta'}J$ at $\beta' = \widetilde{\beta}'_k$. Substituting Equation (14) into Equation (15), Equation (15) is simplified as:

$$\widetilde{\beta}'_{k+1} = \widetilde{\beta}'_k - \Delta_{k+1}(\widetilde{\beta}'_k - (2A'^H A' + \lambda pD(\widetilde{\beta}'_k))^{-1}2A'^H b) \tag{16}$$

Through the above derivation, 3-D sparse imaging problem is solved by iterating Equation (16). In order to make the iteration converge quickly, variable step $\Delta_{k+1} = (\Delta_k)^{0.9}$ is adopted in the iteration. In addition, Bayesian information criteria [25] is used to determine the shrinkage parameter $p$, and the regularization parameter $\lambda$ is selected by the method proposed in [26].

Crucially, an optimized signal processing scheme is utilized in the proposed algorithm. As seen from Equation (16), $A'^H A'$ and $A'^H b$, which are independent of $\widetilde{\beta}'_k$, do not need to be iterated in each iteration. In addition, the dimension of $A'^H A'$ and $A'^H b$, which are $N1 \times N1$ and $N1 \times 1$, respectively, are too much smaller than that of $A$, thus the calculation and storage of $A'^H A'$ and $A'^H b$ are feasible. Therefore, $A'^H A'$ and $A'^H b$ are calculated and stored before the iteration of the algorithm. According to the obtained $A'^H A'$ and $A'^H b$, the iteration of Equation (16) can be performed efficiently. This optimized signal processing scheme reduces the compactional and storage pressure largely.

Based on the above ideas, the difficulty of large-scale computation involved in direct 3-D sparse reconstruction is solved. When $\|\widetilde{\beta}'_{k+1} - \widetilde{\beta}'_k\|_2^2 / \|\widetilde{\beta}'_k\|_2^2$ is less than the preset threshold $\tau$, the iteration exits and we obtain the reconstruction result $\widetilde{\beta}' = \widetilde{\beta}'_{k+1}$. Once we get the reconstruction result $\widetilde{\beta}'$, the 3-D imaging result $\widetilde{\beta}$ can be obtained by:

$$\widetilde{\beta} = \begin{cases} \widetilde{\beta}' & (x, y, z) \in C' \\ 0 & others \end{cases} \tag{17}$$

The specific steps of the proposed sparse imaging algorithm are summed up in Table 1.

**Table 1.** The steps of the proposed sparse imaging algorithm.

---

**A Direct 3-D Sparse Imaging Algorithm for 3-D SAR using Non-uniform Samples**

---

1. Imaging by $3D - NUFFT$ to get the initial imaging result $\widehat{\beta}$.

2. According to Equation (8) and the obtained $\widehat{\beta}$, determine the target area $C'$.

3. Divide the aperture into several sub-apertures.

4. **for each** sub-aperture **do**
5.     According to the target area $C'$, k-space measurements $G(k_x, k_y, k_z)$ and k-space samples $k_x^{j,q}, k_y^{j,q}, k_z^{j,q}$, calculate and store $A'^H b$ and $A'^H A'$.
6.     Determine the value of the shrinkage parameter $p$ [25] and the regularization parameter $\lambda$ [26].
7.     Calculate $\widetilde{\beta}'_{k+1}$ iteratively according to Equation (16), and get the solution $\widetilde{\beta}' = \widetilde{\beta}'_{k+1}$ once $\|\widetilde{\beta}'_{k+1} - \widetilde{\beta}'_{k}\|_2^2 / \|\widetilde{\beta}'_{k}\|_2^2 < \tau$.
8.     According to Equation (17), splice the complete result $\widetilde{\beta}$ from $\widetilde{\beta}'$.
9. **end for**

10. Fuse the imaging results of all sub-apertures using the generalized likelihood ratio test approach [27] and obtain the final imaging results.
11. **return** the final imaging results.

---

In the 3rd step of the proposed algorithm shown in Table 1, the aperture division is carried out. The 3-D SAR using non-uniform samples shown in Figure 1 has an azimuth range of 50 degrees. For such a wide azimuth range, the isotropic point scattering assumption does not usually hold [28] and we need to divide the aperture into sub-apertures firstly before imaging. After imaging on each sub-aperture, the full-aperture imaging result is obtained by fusing the sub-aperture images, as shown in step 5 in Table 1. For the reasons that the direct 3-D sparse imaging modeling avoids the data interpolation and the proposed algorithm solves the difficulty of large-scale computation involved in direct 3-D sparse reconstruction, 3-D sparse imaging results with guaranteed quality can be obtained by the DTDSI method.

## 3. Experiments and Results

### 3.1. Single Target

For the case of single target, a trihedral is used (see the 3-D CAD model in Figure 3). The center frequency of radar and the bandwidth are 10 GHz and 2 GHz, respectively. The data are generated by CST Microwave Studio software.

In order to verify the DTDSI method for 3-D SAR using non-uniform samples, three different sampling modes are employed to generate the echoes of the single trihedral. The scanning trajectories are shown in Figure 4. Please note that although the three scanning trajectories are different, the azimuth and elevation range of the three flights are both in the range of [66°, 114°] and [18°, 42°], respectively. Therefore, it is reasonable to compare their imaging results. Before imaging, the large aperture of each flight is divided into sub-apertures. Since the azimuth aperture size is 48° for each flight, the apertures of these three flights are both divided into 9 sub-apertures with equal interval, thus the size of each sub-aperture is 9.6° and the overlapping between each sub-aperture is 4.8°. As shown in Figure 4, the number 1 to 9 represent the sequence number of the current sub-aperture. Meanwhile, it is clear that the k-space samples of these three flights are both non-uniform and sparse (see Figure 4b,d,f),

whereas the distributions of the k-space samples of each flight are quite different. Apparently, the flight 1 is the sparsest while the flight 3 is the densest.

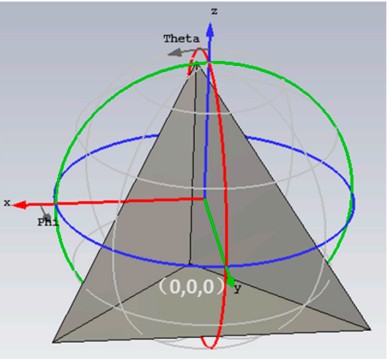

**Figure 3.** 3-D CAD model of the trihedral.

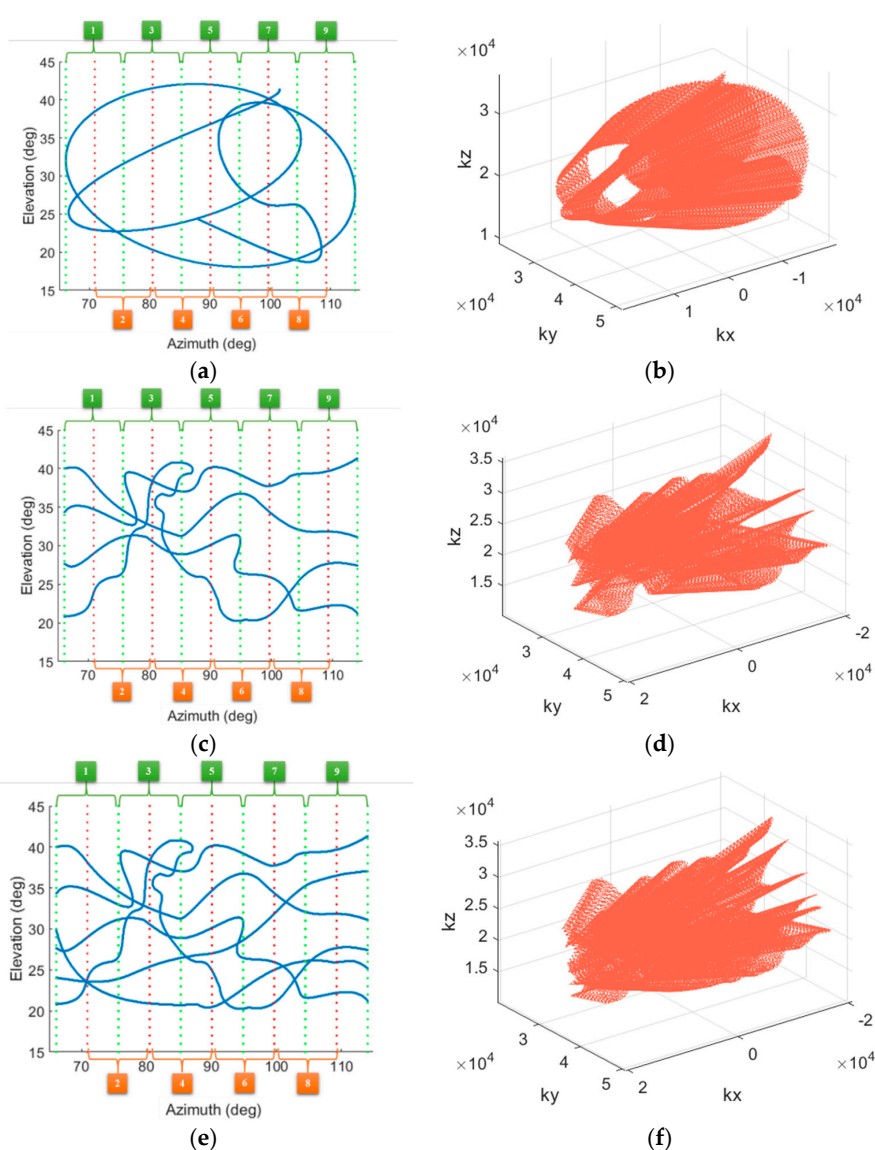

**Figure 4.** The three sampling modes. (**a,c,e**) show scanning trajectories along the elevation and the azimuth of flight 1, flight 2 and flight 3, respectively. (**b,d,f**) show k-space samples of flight 1, flight 2 and flight 3, respectively.

We report the imaging results of the three cases obtained by 3-D NUFFT, Austin's and the DTDSI methods (see Figure 5). The amplitude threshold for Austin's and the DTDSI methods are set to be more than −20 dB to display the imaging results, whereas it is set as −10 dB for 3-D NUFFT due to the high side-lobes in the imaging results of 3-D NUFFT. In each sub-image of Figure 5, the cyan part in the middle is the 3D imaging result of the target, and the gray parts around are the 2-D projection results in X-Y, Y-Z and X-Z directions, respectively. It can be seen from Figure 5a,d,e that the imaging results of 3-D NUFFT are different for different sampling modes and they all are unsatisfactory with high side-lobes. However, it is clear that both Austin's and the DTDSI methods can successfully achieve the imaging of the single trihedral for all sampling modes. In addition, the imaging results by the DTDSI method (see each ellipsoid in Figure 5c,f,i) are significantly smaller than those obtained by Austin's method (see each ellipsoid in Figure 5b,e,h), validating that the imaging results of the DTDSI method are sparser and have better resolution.

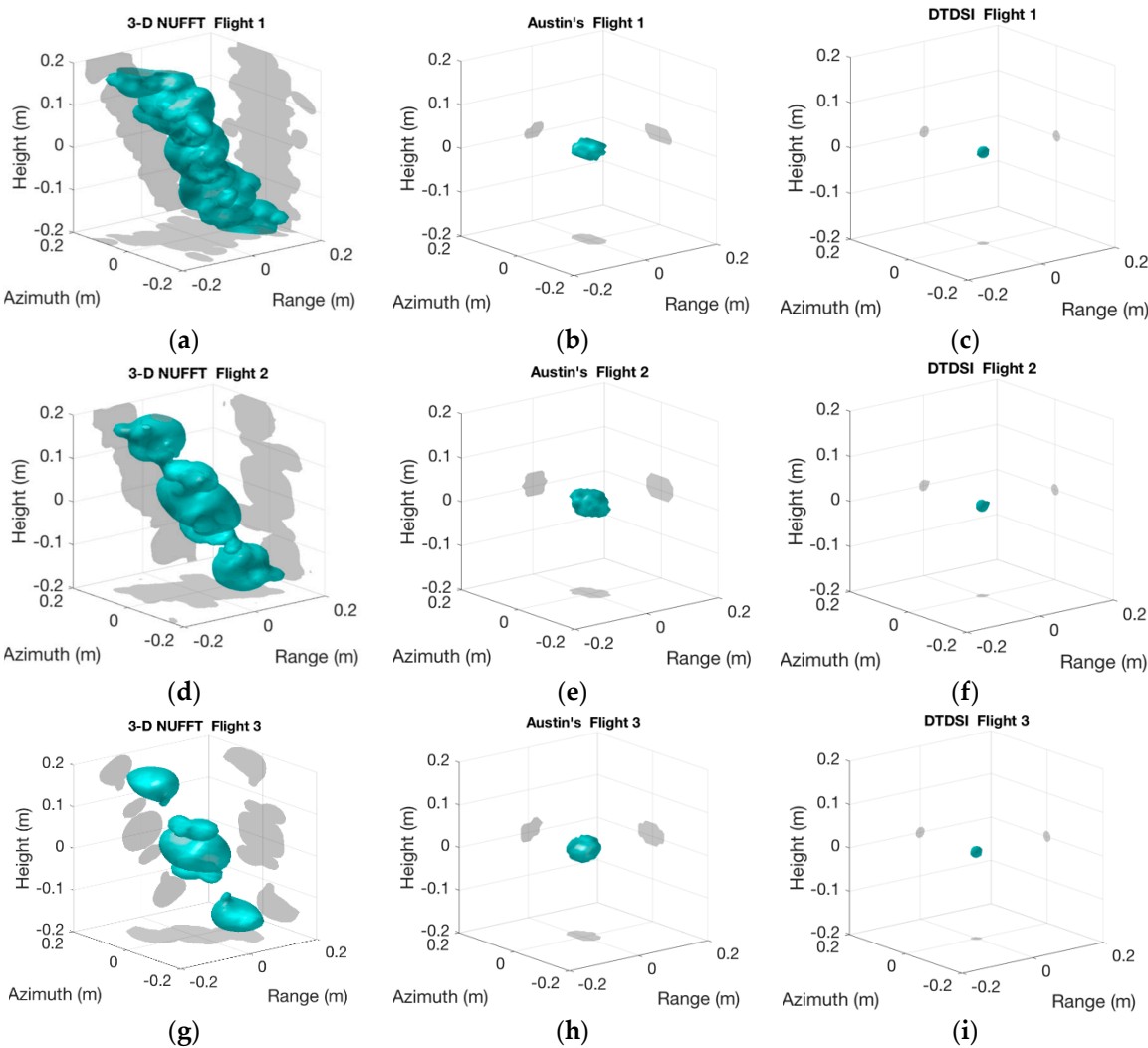

**Figure 5.** 3-D imaging results of the trihedral. (**a**–**c**) show imaging results of flight 1 using 3-D NUFFT (3-D non-uniform fast Fourier transform), Austin's and the DTDSI (direct 3-D sparse imaging) methods, respectively. (**d**–**f**) show imaging results of flight 2 using 3-D NUFFT, Austin's and the DTDSI methods, respectively. (**g**–**i**) show imaging results of flight 3 using 3-D NUFFT, Austin's and the DTDSI methods, respectively.

In order to make a further comparison between Austin's and the DTDSI methods. Figure 6 shows the resolution comparison under different flights and different SNRs. We take the volume of the

imaging result larger than −6 dB as the value of the resolution. Different SNRs ranging from −30 to 30 dB are tested with 50 trials carried out at each level. When the SNR is greater than −25 dB, it can be seen from Figure 6 that all the resolutions are basically stable, and the resolution of the imaging results obtained by the DTDSI method is always better than that by Austin's method for all three sampling modes. Meanwhile, the resolution of the DTDSI method is basically the same for different sampling modes, whereas the resolution of the Austin's method for three sampling modes is different. When the SNR is less than −25 dB, the imaging results of the two methods are getting worse with the decrease of the SNR. In summary, Austin's and the DTDSI methods basically have the same anti-noise performance. The resolution of the imaging result obtained by the DTDSI method is always better than by Austin's method, no matter which sampling mode is utilized.

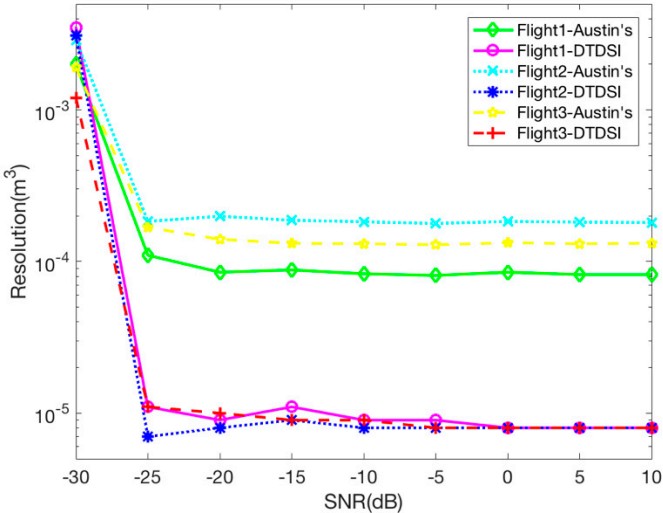

**Figure 6.** Resolution comparison of Austin's and the DTDSI methods under different flights and different SNRs.

Table 2 shows the computational time of 3-D NUFFT, Austin's and the DTDSI methods. It can be seen that 3-D NUFFT requires the lowest time, whereas the Austin's method takes more time than 3-D NUFFT due to the interpolation by 3-D NUFFT is included. Since the data are non-uniform, it is reasonable that the convergence of the DTDSI method requires more time than the Austin's method. In summary, the computational time of the DTDSI method is the same order of magnitude as the other methods.

**Table 2.** Computational time of different imaging methods.

| Method | 3-D NUFFT | Austin's | DTDSI |
|---|---|---|---|
| Time (s) | 6.48 | 9.66 | 15.11 |

### 3.2. Multi Targets

For the case of multi targets, two adjacent trihedrals are used (see the 3-D CAD model in Figure 7). The center frequency of the radar is 10 GHz, and the bandwidth is 2 GHz. The data are generated by CST Microwave Studio software. The scanning trajectory of radar along the elevation and the azimuth is consistent with that of flight 1 shown in Figure 4a.

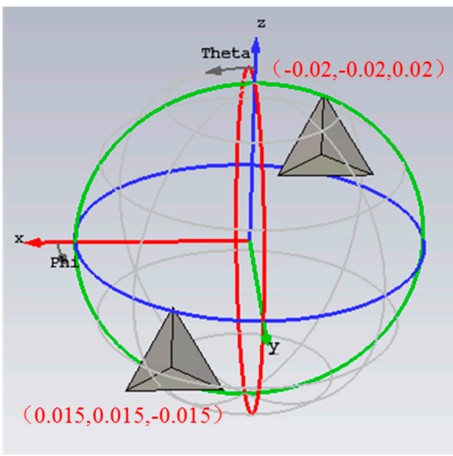

**Figure 7.** 3-D CAD model of two adjacent trihedrals.

The 3-D imaging results of two adjacent trihedrals are given in Figure 8. Figure 8a–c show the final imaging results obtained by 3-D NUFFT, Austin's and the DTDSI methods, respectively.

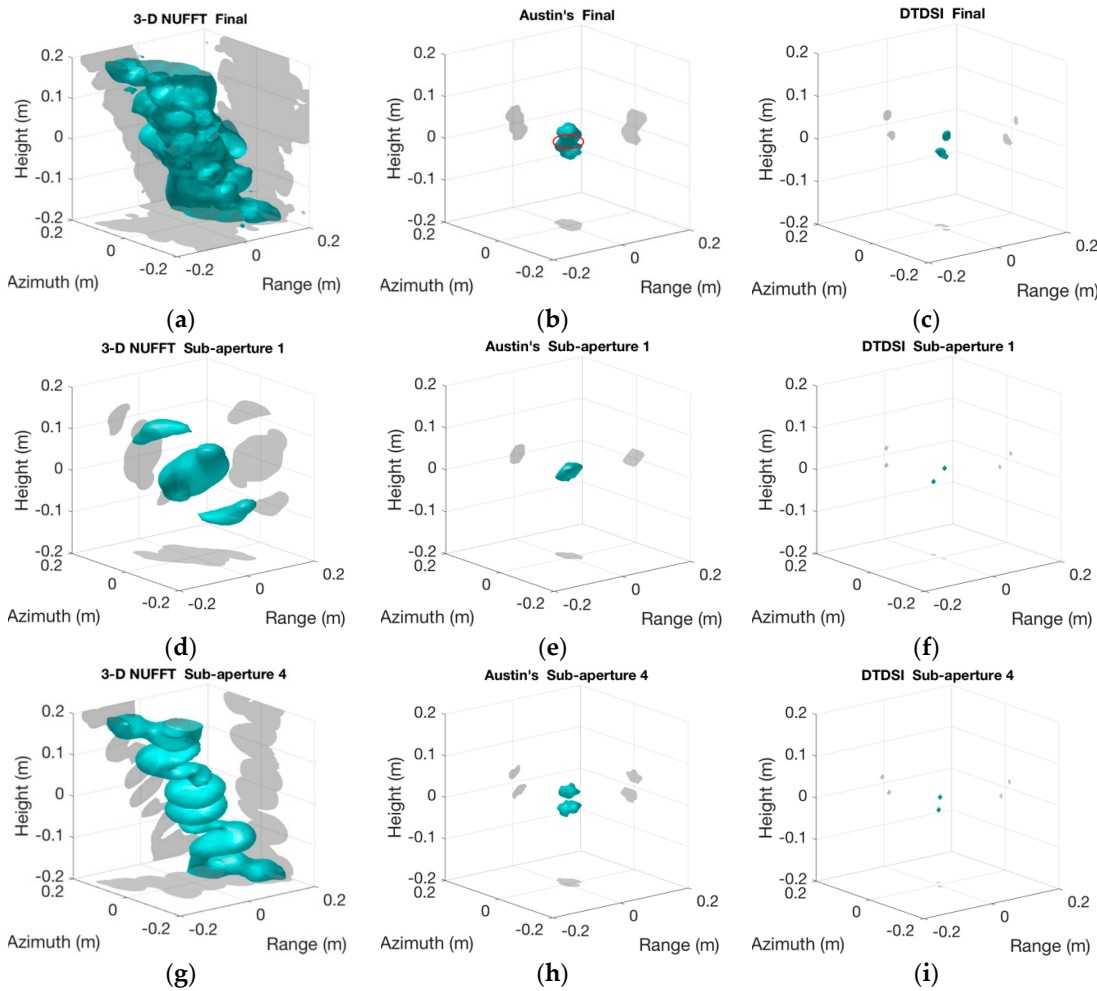

**Figure 8.** 3-D imaging results of two adjacent trihedrals. (**a**,**c**) show final imaging results using 3-D NUFFT, Austin's and the DTDSI methods, respectively. (**d**–**f**) show imaging results of sub-aperture 1 using 3-D NUFFT, Austin's and the DTDSI methods, respectively. (**g**–**i**) show imaging results of sub-aperture 4 using 3-D NUFFT, Austin's and the DTDSI methods, respectively.

As can be seen in Figure 8a, the imaging result obtained by 3-D NUFFT is poor and has high side-lobes. Austin's and the DTDSI methods reduce the side-lobes and obtain the sparse imaging result (see Figure 8b,c). However, the imaging result obtained by Austin's method does not match with the real scene (see Figure 8b), whereas the DTDSI method successfully distinguishes two adjacent trihedrals (see Figure 8c). For further comparative analysis, the imaging result of sub-aperture 1 and 4 by different imaging methods are also displayed in Figure 8. It can be seen from Figure 8f,i that the DTDSI method successfully distinguishes two adjacent trihedrals both in sub-aperture 1 and 4. However, in sub-aperture 1, the imaging result obtained by Austin's method does not match with the real scene, since the cyan part in Figure 8e is not separated by two parts. Similarly, the imaging result obtained by 3-D NUFFT does not distinguish two adjacent trihedrals in sub-aperture 1 (see Figure 8d). In fact, since the data interpolation by 3-D NUFFT is included in Austin's method, the performance of Austin's method is basically determined by 3-D NUFFT. For sub-aperture 1, the interpolated data bring extra errors, thus the imaging result of 3-D NUFFT and Austin's methods are affected badly. Because the imaging results in sub-aperture 1 are not good, the final imaging results shown in Figure 8a,b are poor. In contrast, the DTDSI method does not require data interpolation, thus the authenticity of data is guaranteed and the imaging results are more accurate (see Figure 8c,f,i).

Furthermore, according to the known position of the two adjacent trihedrals, Table 3 gives the Mean Squared Error (MSE) of the imaging results when using different imaging methods. It can be seen that the MSE of 3-D NUFFT is too huge, which is caused by the high side-lobes in Figure 8a. Comparing with the method of Austin, the MSE of the DTDSI method is obviously smaller. In fact, the cyan part in Figure 8b is supposed to be two separated parts, while the extra nonzero part in the middle, which is shown by the red circle in Figure 8b, increases the MSE largely. In contrast, the imaging result shown in Figure 8c consists of two separated parts, matching with the ground truth. Therefore, the imaging result obtained by the DTDSI method is more accurate and has a higher resolution, compared with the other two methods.

**Table 3.** The MSE of the two adjacent trihedrals when using different imaging methods.

| Method | 3-D NUFFT | Austin's | DTDSI |
|--------|-----------|----------|-------|
| MSE | 78.35 | 1.70 | 0.16 |

In the third experiment, thirteen trihedrals which form a shape of an aircraft is placed in the scene, and its 3-D CAD model is shown in Figure 9. The length and width of the aircraft are 0.7 m and 0.6 m, respectively. The tilt angle of the aircraft in the scene is 30 degrees, thus the hight of the imaging scene is 0.35 m. The center frequency of radar and the bandwidth are 10 GHz and 2 GHz, respectively. The data are generated by CST Microwave Studio software. The scanning trajectory of radar along the elevation and the azimuth is consistent with that of flight 1 shown in Figure 4a.

Figure 10 depicts the 3-D imaging results of the aircraft using different imaging methods. As can be seen in Figure 10a, the imaging result obtained by 3-D NUFFT has high side-lobes and the outline of the aircraft cannot be seen from the imaging result. The imaging result obtained by Austin's method reduces most of the side-lobes and gives the rough outline of the aircraft (see Figure 10b). However, there are still some side-lobes in the imaging result, which are highlighted by orange circles in Figure 10b. In addition, the red circle in Figure 10b indicates that the two adjacent trihedrals on the aircraft's wing are not distinguished. As seen from Figure 10c, the imaging result obtained by the DTDSI method matches with the real scene. The outline of the aircraft is very clear and the thirteen trihedrals are all well imaged in Figure 10c, since the side-lobes are removed fully by the DTDSI method.

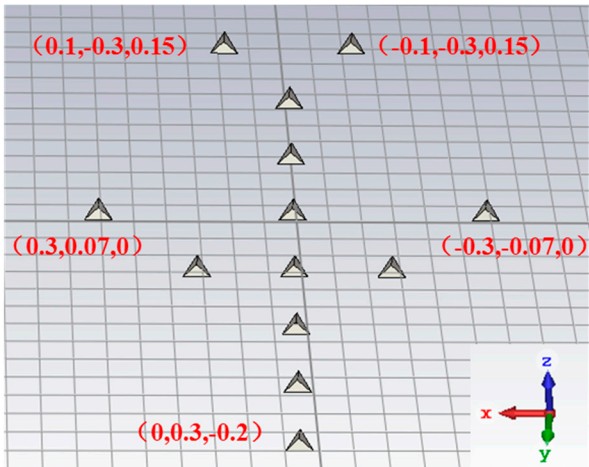

**Figure 9.** 3-D CAD model of an aircraft.

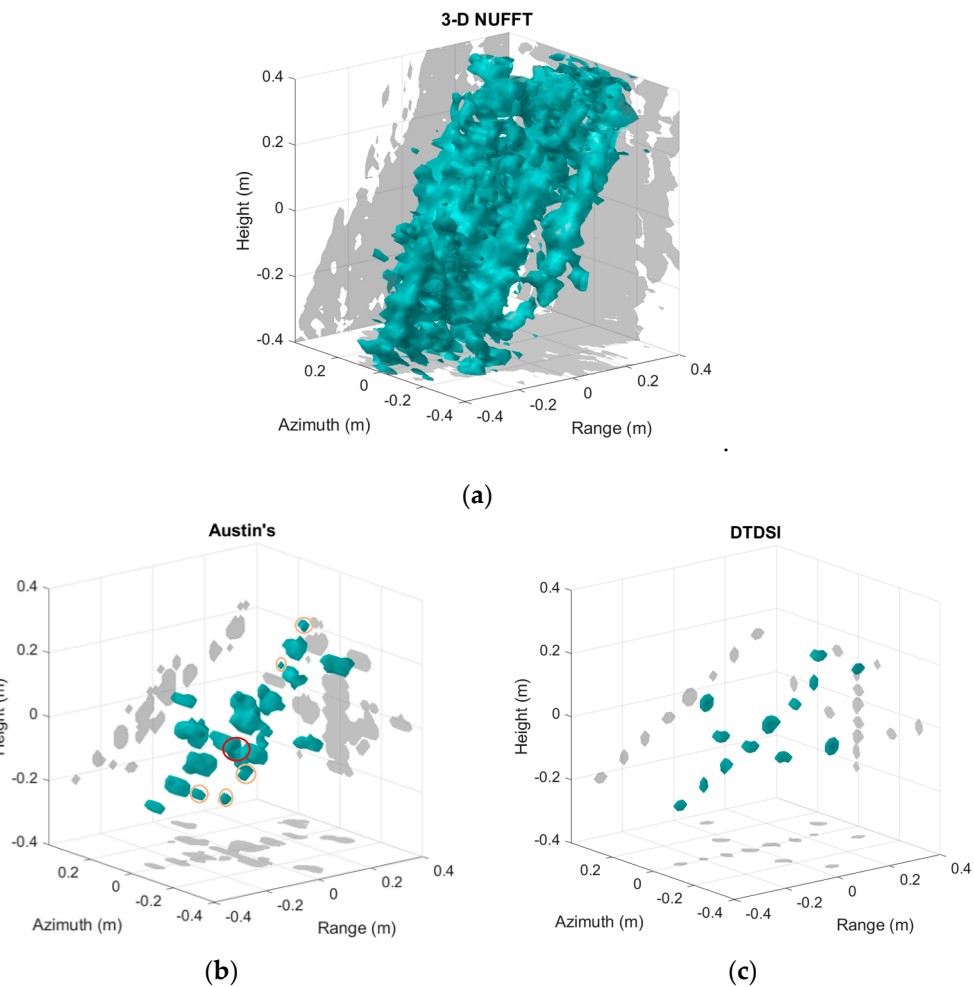

**Figure 10.** 3-D imaging results of the aircraft using different imaging methods. (**a**) 3-D NUFFT. (**b**) Austin's. (**c**) DTDSI.

Furthermore, according to the known position of the aircraft, Table 4 gives the MSE of the imaging results when using different imaging methods. Because the high side-lobes shown in Figure 10a, the MSE of 3-D NUFFT is much higher than the other two methods. Comparing with the method of Austin, the MSE of the DTDSI method is too much smaller. Actually, the errors of imaging result

obtained by Austin's method, which are circled in red and orange in Figure 10b, increase the MSE largely. In contrast, the imaging result shown in Figure 10c matches with the real scene well. Therefore, the imaging result obtained by the DTDSI method is more accurate and sparser, compared with the other two methods.

**Table 4.** The MSE of the aircraft when using different imaging methods.

| Method | 3-D NUFFT | Austin's | DTDSI |
|--------|-----------|----------|-------|
| MSE | 119.92 | 3.11 | 0.11 |

In summary, the theoretical analysis and simulation results verify the feasibility and effectiveness of the DTDSI sparse imaging method.

## 4. Conclusions

In this paper, we propose the DTDSI method using non-uniform samples without data interpolation. The imaging problem is directly formulated as a joint sparse reconstruction problem from non-uniform data in 3-D space, thus the authenticity of the data is guaranteed. Based on the dictionary reduction and the optimized signal processing scheme, the proposed algorithm overcomes the difficulty of large-scale computation involved in direct 3-D sparse reconstruction, thus the 3-D sparse imaging can be carried out with the guaranteed performance. Comparing with other imaging methods, the DTDSI method has better resolving ability, lower side-lobes, higher accuracy, and similar computational complexity.

3-D sparse imaging on each sub-aperture independently ignores the correlation between sub-apertures and can hardly guarantee the consistency of scattering among sub-apertures. To make full use of abundant complementary information among sub-apertures, future work will focus on multiple sub-apertures joint 3-D sparse imaging.

**Author Contributions:** D.S. and S.X. designed the algorithm; D.S. and Y.L. performed the algorithm; S.X. and B.P. proposed important modified scheme of the paper; D.S. wrote the paper; X.W. revised the paper. All authors have read and agreed to the published version of the manuscript.

**Funding:** This research was funded by the National Natural Science Foundation of China (No. 61490692, 61490693, 61901499).

**Acknowledgments:** The authors wish to extend their sincere thanks to Yi Zhang (yzhang120@e.ntu.edu.sg) for her careful reading and fruitful suggestions.

**Conflicts of Interest:** The authors declare no conflict of interest.

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
