# Peer review of "Direct 3-D Sparse Imaging Using Non-Uniform Samples Without Data Interpolation"

_electronics, doi:10.3390/electronics9020321_

Round 1

Reviewer 1 Report

Review of “Direct 3-D Sparse Imaging Using Non-uniform Samples without Data Interpolation”

This paper proposed a direct method to obtain high-resolution 3D images without data interpolation. The overall effort is interesting.  But the writing needs to be improved a lot. Many grammar errors and poor writings made the paper hard to follow. Please follow the manicurist template offered by the journal, to re-construct the whole paper.

The following comments need to be adjusted before acceptance to publishing.

comments:

Line 32. Please write more details of the SAR system if the author wants to emphasize the SAR system is drone-based or hand-based which is labor-intensive. Otherwise, multiple measurements do give good results.

Line 34 ‘more fundamental’ than what types of methods?

Line 36  “a novel technology” is better than ‘an emerging  technique’

Line 40  “much sparser than”

Line 40 What’s the figure 1 results come from? Add citation

Line 43 “it is possible to collect non-uniform samples with low cost for several practical applications, i.e. military surveillance” conflicts with your previous statements as “In addition, acquiring uniform and dense samples is impractical in several applications, such as military surveillance”  

Line 44 The statements of Tomo-SAR is very unclear, your citation of Tomo-SAR is a forest-based study, what’s the aim for your proposed study? Forest? Or other? Please re-write the sentence.

Line 47 to Line 54. These sentences are very confusing to me. What’s the propose of your study? Why you do your research? Why “However, the high side-lobes in imaging result brought by 3-D NUFFT has not been studied in this paper.” In half of the paragraph, do you mean there is no relationship from Line 47 to Line 54 with your research? Please re-write the paragraph.

Line 55 I don’t get why “is expected to ….. ” you need to mention more details to lead the statement.

Line 47-Line 64. Again, the logical flow in this whole paragraph is a mess. There are 3-4 “However” make the entire paragraph extremely hard to follow. Re-write the whole paragraph.

Line 68-Line 84. This is a “what I am going to do” section in the introduction. Please cut the paragraph to 6-7 lines. Also, Line 81 to Line 84 are just repeated the subtitles, remove them.

Please put section 2, 3, and 4 into one section as “Methods and Data”

Line 97 “where j denotes the index of the frequency f and q denotes the index of the observation angle” is unclear

Line 105-Line 113. These statements read like the introduction section.

Line 124-Line 134. These statements also belong to the Introduction. The paper makes me very confusing, why insert a large amount of introduction within the equations (method)? Both the introduction and method need to be re-arranged.

Line 153 to Line 224 OK, This section reads like the actual Method section. My advice is to remove irrelative equations from the paper (there are a lot). And try to make your idea clear (e.g., why I need to do in Introduction, what I did in Method, and what I got from my new method in Results). Try to avoid telling readers “why” by using many equations. Try to use your words to summary it.

Line 225 Remove sentences from Line 226 to Line 229. Put these details to Line 78

Line 225 Give your method a name. ‘Proposed’ is ‘some method I want to do, but I don’t know if it is a good method yet’ It is also good for readers in the comparison section.

Figure 5 label different methods in the figure and the captions. Similar to the following figures.

Figure 6 is hard to separate different lines. Please re-plot

Table 2 why the Proposed method used more times but the author claimed it is a time efficacy method in the introduction.

Line 342, spell out the full name of MSE

Table 4. I would like to see more other error metrics than MSE, e.g., RMSE, R2, mean normalized bias, absolute mean error.

Author Response

Response to Reviewer 1 Comments

Point 1: This paper proposed a direct method to obtain high-resolution 3D images without data interpolation. The overall effort is interesting.  But the writing needs to be improved a lot. Many grammar errors and poor writings made the paper hard to follow. Please follow the manicurist template offered by the journal, to re-construct the whole paper

Response 1: We feel great thanks for your professional comments on our work. According your comments and the manicurist template offered by the journal, we have made several changes in the revised manuscript for better understanding. The modified parts are highlighted in red. And the details are listed in cover letter.

Point 2:Line 32. Please write more details of the SAR system if the author wants to emphasize the SAR system is drone-based or hand-based which is labor-intensive. Otherwise, multiple measurements do give good results.

Response 2:Thanks for your great suggestion. Based on your comments, we have already added more details of the SAR system in our revised manuscript.

Specifically, the statement of “However, uniform and dense samples can be obtained only when the targets are observed by the SAR system several times, thus making data collection time-consuming [7,8].” is replaced by “Generating high-resolution 3-D images using uniform and dense samples requires that the data be collected over a densely sampled set of points in both azimuth and elevation angle [7], as shown in the cyan dotted lines representing a scanning trajectory of Tomo-SAR along the elevation and the azimuth in Figure 1. Collecting data from many closely spaced linear flight passes need the targets be observed by the SAR system several times, thus making data collection time-consuming and high-cost [7,8].”.

Point 3:Line 34 ‘more fundamental’ than what types of methods?

Response 3:Thanks for your comments. Actually, “more fundamental” means that “the imaging using non-uniform samples” is more fundamental than “the imaging using uniform and dense samples”. We have already rewritten the relevant sentences in our revised manuscript to make it clear.

Specifically, the statement of “Consequently, the imaging methods for non-uniform samples are more fundamental and critical in practical applications.” is replaced by “Consequently, there is motivation to consider the imaging using non-uniform samples, which are more fundamental and critical than uniform and dense samples in practical applications.”.

Point 4:Line 36  “a novel technology” is better than ‘an emerging  technique’

Response 4:Thanks for your great suggestion. In our revised manuscript, “an emerging technique” is changed to “a novel technology”.

Point 5:Line 40  “much sparser than”

Response 5:Thanks for your careful checks. In our revised manuscript, “sparser much than” is corrected as “much sparser than”.

Point 6:Line 40 What’s the figure 1 results come from? Add citation

Response 6:Thanks for your comments. The results of Figure 1 come from Reference [7]. Based on your suggestion, we have already added citation of Figure 1 in our revised manuscript.

The statement of “The curve trajectory in Reference [7] is used as the trajectories for UAVs. As shown in Figure 1, the green, red, blue, yellow and black curves represent the scanning trajectory of five UAVs along non-linear flight paths, respectively.” is added in Line 44.

Austin, C.D.; Ertin, E.; Moses, R.L. Sparse signal methods for 3-D radar imaging. IEEE J. Sel. Top. Signal Process.2011, 5, 408–423.

Point 7:Line 43 “it is possible to collect non-uniform samples with low cost for several practical applications, i.e. military surveillance” conflicts with your previous statements as “In addition, acquiring uniform and dense samples is impractical in several applications, such as military surveillance”

Response 7:Thanks for your comments. Based on your comments, we have already rewritten the relevant part in our revised manuscript to make it clear.

Specifically, the statement of “it is possible to collect non-uniform samples with low cost for several practical applications, i.e. military surveillance.” is replaced by “collecting non-uniform samples saves cost greatly and is feasible for several practical applications, i.e. military surveillance.”.

Point 8:Line 44 The statements of Tomo-SAR is very unclear, your citation of Tomo-SAR is a forest-based study, what’s the aim for your proposed study? Forest? Or other? Please re-write the sentence.

Response 8:Thanks for your careful checks. We are sorry for our carelessness. The citation [15] is not suitable here. Actually, our proposed study aims at 3-D imaging for non-uniform samples. And the imaging targets for our study are not forest, but some man-made objects, as the abstract described “especially the targets consists of isolated scaterers”. We have already rewritten the sentence.

In our revised manuscript, the statement of “traditional 3-D imaging methods, which usually aim at the uniform and dense samples such as Tomo-SAR [15,16], can hardly be suitable for 3-D SAR using non-uniform samples.” is replaced by “traditional 3-D imaging methods [5,15,16], which are generally used to solve the imaging problem for uniform and dense samples, can hardly be suitable for 3-D SAR using non-uniform samples. The imaging for non-uniform samples, what we pay attention on in this paper, is very different from the imaging for uniform and dense samples.”.

Feng, D.; An, D.; Huang, X.; Li, Y. A Phase Calibration Method Based on Phase Gradient Autofocus for Airborne Holographic SAR Imaging. IEEE Geosci. Remote Sens. Lett.2019, PP, 1–5. Xing, S.; Member, S.; Li, Y.; Dai, D.; Wang, X. Three-Dimensional Reconstruction of Man-Made Objects Using Polarimetric Tomographic SAR. IEEE Trans. Geosci. Remote Sens.2013, 51, 3694–3705. Peng, X.; Hong, W.; Wang, Y.; Tan, W.; Wu, Y. Polar Format Imaging Algorithm With Wave-Front Curvature Phase Error Compensation for Airborne DLSLA Three-Dimensional SAR. IEEE Geosci. Remote Sens. Lett.2014,11, 1036–1040.

The References[5,15,16]are all focusing onthe imaging problem for uniform and dense samples. And the imaging targets inReferences[5,15] are some man-made objects.

Point 9:Line 47 to Line 54. These sentences are very confusing to me. What’s the propose of your study? Why you do your research? Why “However, the high side-lobes in imaging result brought by 3-D NUFFT has not been studied in this paper.” In half of the paragraph, do you mean there is no relationship from Line 47 to Line 54 with your research? Please re-write the paragraph.

Response 9:Thanks for your comments. The propose of our study is to improve the quality of the imaging results when using non-uniform samples. Since the existing 3-D imaging method for non-uniform samples are based on data interpolation by local information, the accuracy of the imaging result may be affected. Therefore, in order to avoid data interpolation, we propose the direct 3-D sparse imaging method in this paper. The difficulty of large-scale computation involved in direct 3-D sparse reconstruction is solved by our proposed method, and the accuracy imaging result can be obtained meanwhile.

Based on your comments, the statement of “However, the high side-lobes in imaging result brought by 3-D NUFFT has not been studied in this paper.” is deleted in the revised manuscript. In addition, we have already rewritten this part in the revised manuscript.

Point 10:Line 55 I don’t get why “is expected to ….. ” you need to mention more details to lead the statement.

Response 10:Thanks for your suggestion. Actually, in sparse reconstruction, regularization enforcing sparsity to obtain sparse results. Combining the sparse reconstruction with the sparsity of the data to carry out sparse imaging can generate sparse results. Compared with traditional imaging method, results obtained by sparse imaging are sparse, that is, have few side-lobes. Based on your comments, we have already rewritten the relevant sentences in our revised manuscript to make it clear.

Specifically, the statement of “Sparse imaging, as a method of model matching, is expected to reduce high side-lobes and obtain high-resolution imaging results [19,20].” is replaced by “Sparse reconstruction is a method of model matching, in which regularization enforcing sparsity to obtain sparse results [19,20]. Combining the sparse reconstruction with the sparsity of the data to carry out 3-D sparse imaging for non-uniform samples is expected to reduce high side-lobes and obtain high-resolution imaging results.”.

Point 11:Line 47-Line 64. Again, the logical flow in this whole paragraph is a mess. There are 3-4 “However” make the entire paragraph extremely hard to follow. Re-write the whole paragraph.

Response 11:Thanks for your suggestion. We have already rewritten the entire paragraph in the revised manuscript for better understanding.

The revised version of this paragraph is listed:

In the literatures of 3-D imaging, there are few published works focusing on non-uniform samples, and the involved difficulty rely on that linear filtering is not applicable for imaging since the k-space samples of 3-D SAR using non-uniform samples is sparse and non-uniform. Specifically, due to the high coupling between the elevation and the azimuth resulting from the non-uniform k-space samples, it is impossible to estimate the height independently after the two-dimensional imaging [17]. Although full 3-D imaging is an advisable way, using Fourier processing methods, such as 3-D non-uniform fast Fourier transform (3-D NUFFT), generates poor imaging results with high side-lobes due to the sparsity of the k-space samples [17,18]. Sparse reconstruction is a method of model matching, in which regularization enforcing sparsity to obtain sparse results [19,20].Combining the sparse reconstruction with the sparsity of the data to carry out 3-D sparse imaging for non-uniform samples is expected to reduce high side-lobes and obtain high-resolution imaging results. However, the large-scale computation involved in 3-D sparse reconstruction makes it a challenging and difficult task [21]. To reduce the computational complexity, Austin proposed a sparse imaging method based on data interpolation in k-space and fast Fourier transform (FFT) operation [7]. After interpolation, the updated dictionary matrix can be replaced by FFT operation for fast calculation in the procedure of sparse reconstruction due to the uniformity of the interpolated data. Although this method solves the difficulty of large-scale computation and makes sparse imaging for non-uniform samples feasible, the data interpolation by local information involves the high potential for data-gridding errors and in turn affect the accuracy of the imaging results.

Point 12:Line 68-Line 84. This is a “what I am going to do” section in the introduction. Please cut the paragraph to 6-7 lines. Also, Line 81 to Line 84 are just repeated the subtitles, remove them.

Response 12:Thanks for your great suggestion. The relevant paragraph has already been cut in the revised manuscript. In addition, Line 81 to Line 84 is deleted.

The revised version of this paragraph is listed:

In order to exploit sparse imaging and avoid the adverse effects caused by local interpolation, the direct 3-D sparse imaging (DTDSI) method is proposed in this paper. Firstly, the imaging problem is directly regarded as a joint sparse reconstruction problem from non-uniform data without data interpolation in 3-D space. To address the difficulty of large-scale computation involved in direct 3-D sparse reconstruction, we then reduce the dimension of the dictionary matrix via candidate scattering centers selection. Finally, combining the Gauss iterative method and the optimized signal processing scheme, an algorithm is proposed to solve the updated sparse imaging model. To evaluate the performance of the DTDSI method, we compare it with 3-D NUFFT and Austin’s method via experiments of electromagnetic simulation data. The experiments are conducted in MATLAB R2016b, and tested on a computer with Intel Core I5-6500 CPU and 12GB RAM.

Point 13:Please put section 2, 3, and 4 into one section as “Methods and Data”

Response 13:Thanks for your suggestion. We have already put section 2, 3, and 4 into one section in the revised manuscript.

Point 14:Line 97 “where j denotes the index of the frequency f and q denotes the index of the observation angle” is unclear

Response 14:Thanks for your comments. We have already rewritten the relevant sentences in our revised manuscript to make it clear.

The revised version of this paragraph is listed:

The support of each k-space measurement is a line segment in k-space samples  with extent  rad/m centered at  rad/m, and oriented at observation angle  determined by the location of the radar. A set of k-space samples indexed on  is given by:

(3)

where the frequency  is sampled as andthe observation angle is sampled as.

Point 15:Line 105-Line 113. These statements read like the introduction section.

Response 15:Thanks for your comments. In our revised manuscript, these statements have already been moved to the introduction section.

The revised version of the relevant paragraph is listed:

The task of this paper is to estimate the reflectivity function  from the known k-space measurements  according to their relationship shown in Equation (2). For the 3-D SAR using non-uniform samples, the scanning trajectory of radar (see Figure 1) is a set of random curves, which leads to the formation of a complex baseline. Consequently, the k-space samples , which are determined by the observation angle, namely, the scanning trajectory, are non-uniform and sparse, as shown in Figure 2. In order to obtain 3-D high-resolution imaging results for the non-uniform and sparse k-space samples,we combine the sparse reconstruction with the sparsity of the data and propose the DTDSI method.

Point 16:Line 124-Line 134. These statements also belong to the Introduction. The paper makes me very confusing, why insert a large amount of introduction within the equations (method)? Both the introduction and method need to be re-arranged.

Response 16:Thanks for your suggestion. In our revised manuscript, these statements have already been moved to the introduction section. Both the introduction and method have already been re-arranged.

The revised version of the relevant paragraph is listed:

In order to avoid errors caused by data interpolation, we abandon data interpolation and directly utilize the dictionary matrix in Equation (5) to build the direct 3-D sparse imaging model.

Point 17:Line 153 to Line 224 OK, This section reads like the actual Method section. My advice is to remove irrelative equations from the paper (there are a lot). And try to make your idea clear (e.g., why I need to do in Introduction, what I did in Method, and what I got from my new method in Results). Try to avoid telling readers “why” by using many equations. Try to use your words to summary it.

Response 17:Thanks for your great suggestion. In our revised manuscript, the irrelative equations (Equation (17) and Equation (18)) have already been removed from the paper. According to your comments, we have already rewritten the relevant sentences to make my idea clear.

Point 18:Line 225 Remove sentences from Line 226 to Line 229. Put these details to Line 78

Response 18:Thanks for your great suggestion. these details have already been moved to introduction in our revised manuscript.

Point 19:Line 225 Give your method a name. ‘Proposed’ is ‘some method I want to do, but I don’t know if it is a good method yet’ It is also good for readers in the comparison section.

Response 19:Thanks for your great suggestion. DTDSI (direct three-dimensional sparse imaging) is used as the name of our proposed method. We have already changed “the proposed method” into “the DTDSI method” in the revised manuscript.

Point 20:Figure 5 label different methods in the figure and the captions. Similar to the following figures.

Response 20:Thanks for your great suggestion. In our revised manuscript, we have already added labels to Figure 5,8,10. The revised version of Figure 10b is:

Point 21:Figure 6 is hard to separate different lines. Please re-plot

Response 21:Thanks for your great suggestion. Figure 6 has already been replotted in the revised manuscript. The revised version of Figure 6 is:

According to the revised version of Figure 6, the statement of “When the SNR is greater than -25 dB, it can be seen from Figure 6 that all the resolutions remain stable, and the resolution of the imaging results obtained by the proposed method is always better than that by Austin’s method for all three sampling modes. Meanwhile, the resolution of the proposed method is the same for different sampling modes, whereas the resolution of the Austin’s method for three sampling modes is different.” is replaced by “When the SNR is greater than -25 dB, it can be seen from Figure 6 that all the resolutions are basically stable, and the resolution of the imaging results obtained by the proposed method is always better than that by Austin’s method for all three sampling modes. Meanwhile, the resolution of the proposed method is basically the same for different sampling modes, whereas the resolution of the Austin’s method for three sampling modes is different.

Point 22:Table 2 why the Proposed method used more times but the author claimed it is a time efficacy method in the introduction.

Response 22:Thanks for your comments. We claimed the proposed method an efficient method because the proposed method overcomes the difficulty of large-scale computation involved in direct 3-D sparse reconstruction. Although the proposed method requires more time than the Austin’s method, the computational time of the proposed method is the same order of magnitude as the other methods. Therefore, “Comparing with other imaging methods, the proposed method has better resolving ability, lower side-lobes, higher accuracy, and similar computational complexity.” is written in introduction. In summary, “efficiency” is not precise. We have already deleted the statement of “efficiency” in the revised manuscript.

Point 23:Line 342, spell out the full name of MSE

Response 23:Thanks for your careful checks. In our revised manuscript, “MSE” is corrected as “Mean Squared Error(MSE)”.

Point 24:Table 4. I would like to see more other error metrics than MSE, e.g., RMSE, R2, mean normalized bias, absolute mean error.

Response 24:Thanks for your comments. For the cases of two adjacent trihedrals and the aircraft, the following Table gives the RMSE of the imaging results when using different imaging methods.

Target

3-D NUFFT

Austin’s

DTDSI

Two adjacent trihedrals

8.8516

1.30

0.4

Aircraft

10.95

1.76

0.33

It can be seen that RMSE of 3-D NUFFT is the largest. Comparing with the method of Austin, RMSE of DTDSI is obviously smaller. The conclusion indicated by RMSE are consistent with MSE.

Reviewer 2 Report

There are some further discussions on the system analysis as following:

- In equation 1, what is center frequency fc? It is not used in the equation.

What is the frequency in here? What is the observation angle?
- It will be good with the explanation of the SAR operation.
- Is there any connection between the received signal and observation angle from equation 1,2 with section 3 and 4.
- What is j and q in equation 5? Does it connect to section 2?

- Line 146 what is G? Does it connection to equation 2?

- Hopefully, the whole analysis has a consequence for full understanding.

Author Response

Response to Reviewer 2 Comments

Point 1: In equation 1, what is center frequency fc? It is not used in the equation.

Response 1: Thanks for your comments.  represents the center frequency of the wideband signal . And the wideband signal  is what radar transmits. Based on your comments, we have revised our manuscript.In our revised manuscript, the statement of “where  is the time,  is the speed of light,  is the known band-limited signal,  represents convolution.” was changed to “where  is the time,  is the speed of light, is the known wideband signal with center frequency  and bandwidth ,  represents convolution.”. In addition, center frequency  is used in the following statements in the revised manuscript.

Point 2:What is the frequency in here? What is the observation angle?- It will be good with the explanation of the SAR operation.

Response 2:Thanks for your great suggestion. Actually,  represents the frequency of the wideband signal  and is determined by center frequency  and bandwidth  of . Since the k-space measurements  are obatined from the known wideband signal  according to the projection slice theorem, there is an relationship between the frequency  and the K-space samples as shown in Equation (3).In addition, the observation angle is the location of the radar. Since the radar is located at azimuth  and elevation , the observation angle is . Based on your comments, we have revised our manuscript. In our revised manuscript, the statement of “where  is the k-space measurements and  represent the k-space samples:” was changed to “where  is the k-space measurements obtained from the received signal  and the wideband signal .The support of each k-space measurement is a line segment in k-space samples  with extent  rad/m centered at  rad/m, and oriented at observation angle  determined by the location of the radar. A set of k-space samples indexed on  is given by:”.

Point 3:Is there any connection between the received signal and observation angle from equation 1,2 with section 3 and 4.

Response 3:Thanks for your professional comments on our work. Yes, there is a connection. Specifically, the imaging method described in Section 3 and 4 are all based on Equation (6) rewritten from Equation (2). In Equation (2), the k-space measurements  are obatined from the received signaland the k-space samples are determined by the observation angle .Therefore, there is aconnection between the received signal and the observation angle from Equation (1)(2) with Section 3 and 4. Based on your comments, we have already rewritten the related part in our revised manuscript to make it clear.

Point 4:What is j and q in equation 5? Does it connect to section 2?

Response 4:Thanks for your comments.  denotes the index of the frequency  and  denotes the index of the observation angle .  and  in Equation (5) are exactly the  and  inSection 2. A set of k-space samples indexed on is given by Equation (3)in section 2.Based on your comments, we have already rewritten the related part in our revised manuscript to make it clear.

Point 5:Line 146 what is G? Does it connection to equation 2?

Response 5:Thanks for your careful checks. G in line 146 does not have any connection to Equation (2). G in line 146 represents Gigabyte. Based on your comments,we have corrected “G” into “GB” in the revised manuscript.

Point 6:Hopefully, the whole analysis has a consequence for full understanding.

Response 6:Thanks for your suggestion. We have made several changes in the revised manuscript for better understanding. The modified parts are highlighted in red. And the details are listed in cover letter.

Reviewer 3 Report

The structure of the article is considered and clear. In the introduction, the background and comprehensive review of the problem's literature were described. The Authors present model of the signal transmitted by the radar far enough from the scene. Direct 3-D Sparse Imaging Modeling and proposed one have been presented. In the manuscript the data generated by CST Microwave Studio software are used. Results of research have been presented in graphic form. Conclusions, on the basis of the research, are clear.

Author Response

Response to Reviewer 3 Comments

Point:The structure of the article is considered and clear. In the introduction, the background and comprehensive review of the problem's literature were described. The Authors present model of the signal transmitted by the radar far enough from the scene. Direct 3-D Sparse Imaging Modeling and proposed one have been presented. In the manuscript the data generated by CST Microwave Studio software are used. Results of research have been presented in graphic form. Conclusions, on the basis of the research, are clear.

Response: We feel great thanks for your professional comments on our work. We have made several changes in the revised manuscript for better understanding. The modified parts are highlighted in red. And the details are listed in cover letter.

Round 2

Reviewer 1 Report

The authors have addressed the comments properly. Accept.

Reviewer 2 Report

Thanks for author's contribution.